American Society for Microbiology | Microbiology Spectrum

# Characterization of *Pseudomonas aeruginosa* L,D-Transpeptidases and Evaluation of Their Role in Peptidoglycan Adaptation to Biofilm Growth

Inès Hugonneau-Beaufet,[a] Jean-Philippe Barnier,[a,b,c] Stanislas Thiriet-Rupert,[d] Sylvie Létoffé,[d] Jean-Luc Mainardi,[a,b,c] Jean-Marc Ghigo,[d] Christophe Beloin,[d] Michel Arthur[a]

[a]Centre de Recherche des Cordeliers, Sorbonne Université, INSERM, Université Paris Cité, Paris, France
[b]Service de Microbiologie, Hôpital Européen Georges Pompidou, AP-HP Assistance Publique-Hôpitaux de Paris, Paris, France
[c]Université Paris Cité, Faculté de Santé, UFR de Médecine, Paris, France
[d]Institut Pasteur, Université Paris Cité, UMR CNRS 6047, Genetics of Biofilms Laboratory, Paris, France

**ABSTRACT** Peptidoglycan is an essential component of the bacterial cell envelope that sustains the turgor pressure of the cytoplasm, determines cell shape, and acts as a scaffold for the anchoring of envelope polymers such as lipoproteins. The final cross-linking step of peptidoglycan polymerization is performed by classical D,D-transpeptidases belonging to the penicillin-binding protein (PBP) family and by L,D-transpeptidases (LDTs), which are dispensable for growth in most bacterial species and whose physiological functions remain elusive. In this study, we investigated the contribution of LDTs to cell envelope synthesis in *Pseudomonas aeruginosa* grown in planktonic and biofilm conditions. We first assigned a function to each of the three *P. aeruginosa* LDTs by gene inactivation in *P. aeruginosa*, heterospecific gene expression in *Escherichia coli*, and, for one of them, direct determination of its enzymatic activity. We found that the three *P. aeruginosa* LDTs catalyze peptidoglycan cross-linking (Ldt$_{Pae1}$), the anchoring of lipoprotein OprI to the peptidoglycan (Ldt$_{Pae2}$), and the hydrolysis of the resulting peptidoglycan-OprI amide bond (Ldt$_{Pae3}$). Construction of a phylogram revealed that LDTs performing each of these three functions in various species cannot be assigned to distinct evolutionary lineages, in contrast to what has been observed with PBPs. We showed that biofilm, but not planktonic bacteria, displayed an increase proportion of peptidoglycan cross-links formed by Ldt$_{Pae1}$ and a greater extent of OprI anchoring to peptidoglycan, which is controlled by Ldt$_{Pae2}$ and Ldt$_{Pae3}$. Consistently, deletion of each of the *ldt* genes impaired biofilm formation and potentiated the bactericidal activity of EDTA. These results indicate that LDTs contribute to the stabilization of the bacterial cell envelope and to the adaptation of peptidoglycan metabolism to growth in biofilm.

**IMPORTANCE** Active-site cysteine LDTs form a functionally heterologous family of enzymes that contribute to the biogenesis of the bacterial cell envelope through formation of peptidoglycan cross-links and through the dynamic anchoring of lipoproteins to peptidoglycan. Here, we report the role of three *P. aeruginosa* LDTs that had not been previously characterized. We show that these enzymes contribute to resistance to the bactericidal activity of EDTA and to the adaptation of cell envelope polymers to conditions that prevail in biofilms. These results indicate that LDTs should be considered putative targets in the development of drug-EDTA associations for the control of biofilm-related infections.

**KEYWORDS** L,D-transpeptidases, biofilms, lipoproteins, peptidoglycan

P eptidoglycan is a major component of the cell envelope that is present in almost all bacterial species. This complex heteropolymer is composed of linear glycan chains cross-linked by short stem peptides to form a three-dimensional network that

Address correspondence to Christophe Beloin, christophe.beloin@pasteur.fr, or Michel Arthur, michel.arthur@crc.jussieu.fr.

The authors declare no conflict of interest.

*[This article was published on 31 May 2023 with Christophe Beloin's name missing from the corresponding author footnote. The footnote was updated in the current version, posted on 5 June 2023.]*

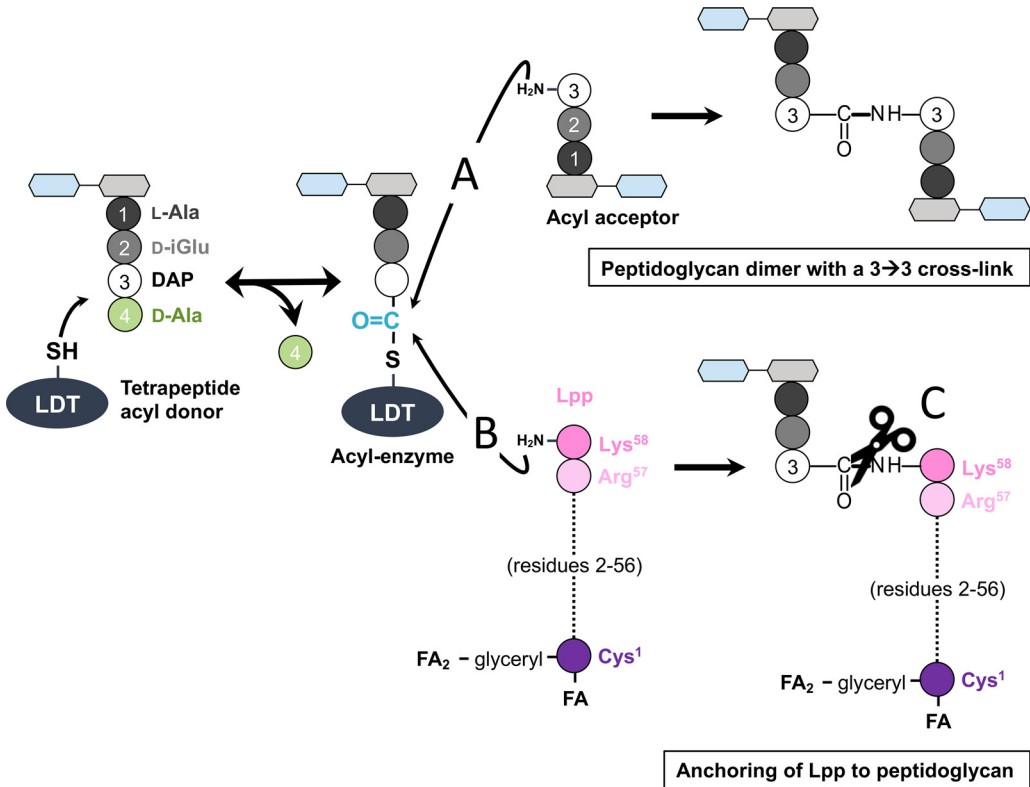

**FIG 1** Two-step reactions catalyzed by LDTs. In the first step, the catalytic cysteine attacks the carbonyl of D-Ala[4] at the extremity of an acyl donor tetrapeptide stem. This results in the formation of an acyl-enzyme and the release of D-Ala[4]. In the second step, the carbonyl of the thioester bond in the acyl-enzyme is attacked by a side chain amine harbored either by DAP[3] in an acceptor stem peptide (reaction A, formation of 3→3 cross-links) or by the C-terminal L-Lys of Lpp (reaction B, Lpp anchoring). Other members of the LDT family hydrolyze the tripeptide→Lpp amide bond (reaction C, release of Lpp). Abbreviations: DAP, diaminopimelic acid; D-iGlu, D-isoglutamic acid; Lpp, major outer membrane Braun lipoprotein; FA, fatty acid.

surrounds the cytoplasmic membrane and provides resistance to the turgor pressure of the cytoplasm (1). The glycan chains are formed by alternating *N*-acetylglucosamine (GlcNAc) and *N*-acetylmuramic acid (MurNAc) residues linked together by $\beta$-1,4 glycosidic bonds. In Gram-negative bacteria, the D-lactoyl group of MurNAc residues are linked to a conserved peptide stem containing an L-alanine at the first position (L-Ala[1]), a D-glutamic acid at the second position (D-Glu[2]), a diaminopimelic acid at the third position (DAP[3]), and two D-alanines at the fourth and fifth positions (D-Ala[4] and D-Ala[5]). In Gram-positive bacteria, DAP[3] is often amidated or replaced by an L-lysine (L-Lys[3]), which can be substituted by a short side chain (e.g., a pentaglycine in *Staphylococcus aureus*).

The peptide stems of two adjacent subunits are linked together by peptide cross-links formed by transpeptidases. Most cross-links connect D-Ala[4] in an acyl donor stem peptide to DAP[3] in an acyl acceptor stem peptide. Synthesis of these so-called 4→3 cross-links is catalyzed by the D,D-transpeptidase activity of penicillin-binding proteins (PBPs) in two steps (2). First, the catalytic serine of PBPs attacks the carbonyl of D-Ala[4] in a donor stem pentapeptide, resulting in the formation of an acyl-enzyme and release of D-Ala[5]. Second, the ester bond of the acyl-enzyme is attacked by the side chain amine of the residue at the 3rd position of an acceptor stem peptide, resulting in the formation of a 4→3 cross-linked peptidoglycan dimer and the release of the PBP (2).

In certain bacteria, peptidoglycan cross-linking is additionally performed by a second family of enzymes, the L,D-transpeptidases (LDTs), which connect residues at the 3rd position of the donor and acceptor stems (3→3 cross-links) (Fig. 1, reaction A) (3, 4). Similar to PBPs, LDTs catalyze peptidoglycan cross-linking by a two-step mechanism involving formation of an acyl-enzyme. However, the acyl donor used by LDTs harbors

a tetrapeptide stem instead of a pentapeptide stem. The tetrapeptide-containing substrate of LDTs is formed by hydrolysis of the D-Ala$^4$–D-Ala$^5$ amide bond of pentapeptide stems by D,D-carboxypeptidases (5–7). LDTs and PBPs also differ by the catalytic nucleophile, Cys versus Ser, and the two enzyme families are structurally unrelated (8). PBPs and LDTs are inactivated through acylation of their nucleophiles. PBPs are potentially inactivated by all classes of $\beta$-lactams. In contrast, LDTs are effectively inactivated only by $\beta$-lactams belonging to the carbapenem class (9–13).

Since the first characterization of an LDT in a highly ampicillin-resistant mutant of *Enterococcus faecium* selected *in vitro* (4), numerous members of the LDT family have been identified in Gram-positive and Gram-negative bacteria as well as in mycobacteria (14–17). The number of LDTs differs between bacterial species, ranging from 1 in *Neisseria meningitidis* and *Helicobacter pylori* to 21 in *Bradyrhizobium japonicum* (18). A wide variety of functions that can be classified in three groups has been associated with these enzymes, as follows.

A first group of LDTs forms 3→3 peptidoglycan cross-links and comprises enzymes with various physiological functions (Fig. 1, reaction A). Certain LDTs participate in the maturation of peptidoglycan in *Mycobacterium smegmatis* (19, 20) and in $\beta$-lactam resistance in mutants of *Escherichia coli* and *E. faecium* selected *in vitro* (4, 7). In these *E. coli* and *E. faecium* mutants, the LDTs can fully replace the PBPs, resulting in a peptidoglycan exclusively containing 3→3 cross-links. The proportion of 3→3 cross-links formed by LDTs is highly variable in wild-type bacteria, ranging from 70% to 80% in *Clostridioides difficile* (14) and mycobacteria (21, 22) to <10% in most species, including in *E. coli* (23). LDTs are essential for virulence in *Mycobacterium tuberculosis* (24) but fully dispensable in *E. coli*, at least for growth in laboratory conditions. In the latter bacterium, the proportion of 3→3 cross-links increases in the stationary phase of growth, but this observation has not been associated with any phenotypic property (25). In *E. coli*, 3→3 cross-link formation was reported to participate in peptidoglycan remodeling, thereby increasing the overall robustness of the bacterial cell envelope in response to defects in the outer membrane (26). In *Salmonella enterica* serovar Typhi, an L,D-transpeptidase plays an essential role in typhoid toxin secretion (27). The enzyme edits the peptidoglycan, i.e., enriches peptidoglycan at the cell poles in 3→3 cross-links, thereby enabling specific cleavage of these cross-links by a specialized muramidase for translocation of the toxin though the peptidoglycan layer. This subsequently enables the release of the toxin through the outer membrane.

A second group of LDTs is specialized in the covalent anchoring of proteins to peptidoglycan (Fig. 1, reaction B). In *E. coli*, three LDTs anchor the Braun lipoprotein (Lpp), providing a link between the peptidoglycan and the outer membrane that is thought to contribute to the stability of the envelope (28–30). In *Coxiella burnetii*, the cell envelope has been proposed to be similarly stabilized by the anchoring of an outer membrane barrel protein to peptidoglycan by an LDT during the stationary phase of growth (31).

A third group of LDTs comprises enzymes acting as hydrolases (Fig. 1, reaction C). In *E. coli*, a member of the LDT family, YafK (also known as DpaA), was found to exclusively display hydrolytic activity for cleavage of the amide bond connecting Lpp to tripeptide stems (32, 33). In combination with LDTs that anchor Lpp, this enzyme may dynamically control the equilibrium between the free and peptidoglycan-linked forms of Lpp.

In addition to the three types of amide bond-forming and -hydrolyzing activities depicted as reactions A, B, and C in Fig. 1, LDTs catalyze the exchange of the terminal D-Ala$^4$ of tetrapeptide stems by glycine and various D-amino (e.g., D-Met) and D-2-hyroxy (e.g., D-lactate) acids (3). For the latter reactions, the acyl-enzyme is attacked by the amine or hydroxyl group of free amino or 2-hydroxy acids. This exchange reaction results in the incorporation of D-amino acids in the peptidoglycan, leading to toxic effects. In *Vibrio cholerae*, noncanonical D-amino acids promote remodeling of peptidoglycan in stationary phase and participate in the control of peptidoglycan abundance and strength (34).

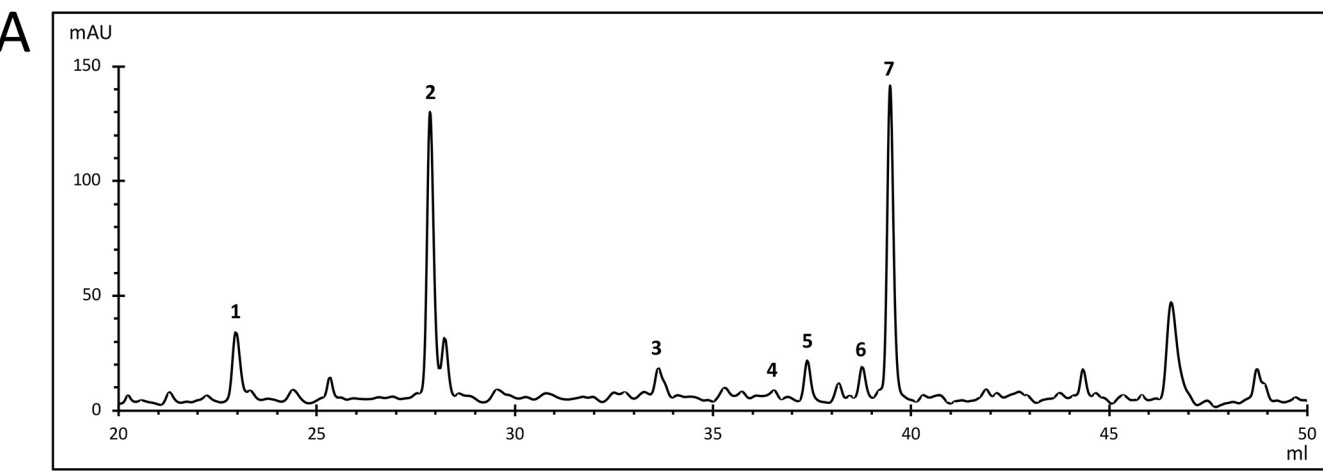

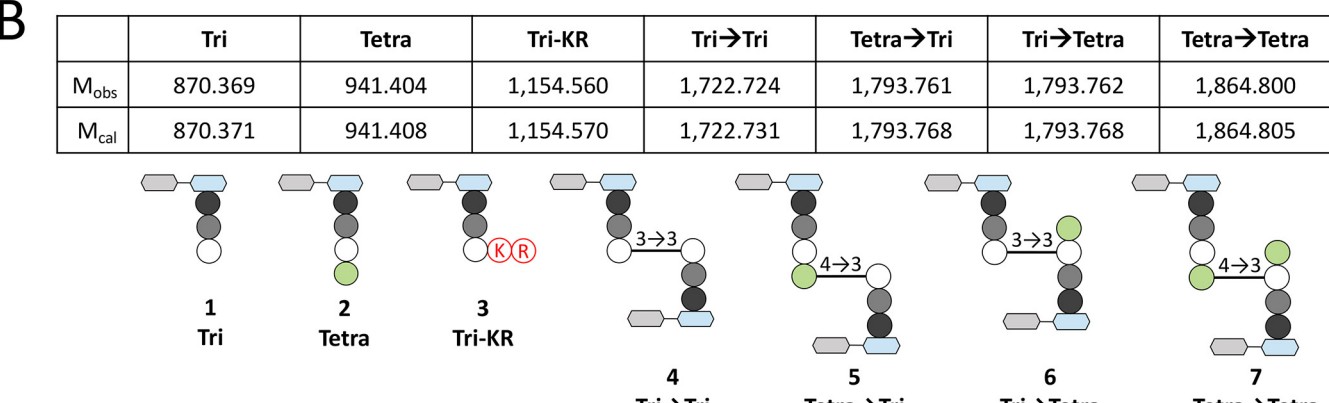

| | Tri | Tetra | Tri-KR | Tri→Tri | Tetra→Tri | Tri→Tetra | Tetra→Tetra |
|---|---|---|---|---|---|---|---|
| $M_{obs}$ | 870.369 | 941.404 | 1,154.560 | 1,722.724 | 1,793.761 | 1,793.762 | 1,864.800 |
| $M_{cal}$ | 870.371 | 941.408 | 1,154.570 | 1,722.731 | 1,793.768 | 1,793.768 | 1,864.805 |

FIG 2 Muropeptide composition of the peptidoglycan of *P. aeruginosa* strain PA14. (A) rpHPLC profile of muropeptides. (B) Structure of muropeptides. $M_{obs}$, observed monoisotopic mass; $M_{cal}$, calculated monoisotopic mass; gray hexagons, GlcNAc; blue hexagons, MurNAc; black circles, L-Ala; gray circles, D-iGlu; white circles, DAP; green circles, D-Ala; K and R circled in red, amino acid residues corresponding to the Arg-Lys C-terminal extremity of OprI.

Attack of the acyl-enzyme by a water molecule results in the hydrolysis of the DAP³–D-Ala⁴ amide bond. In *Acinetobacter baumannii*, a hydrolytic member of the LDT family acts as an L,D-carboxypeptidase, trimming off D-Ala⁴ of tetrapeptide stems and thereby contributing to peptidoglycan recycling (35).

*Pseudomonas aeruginosa* is an opportunistic Gram-negative pathogen, responsible for both acute respiratory infections, in particular ventilator-associated pneumonia, and chronic lung infections in patients with cystic fibrosis and chronic obstructive pulmonary disease. The ability of *P. aeruginosa* to form biofilms is critical in the pathophysiology of these infections (36–38).

In this study, we show that the chromosome of *P. aeruginosa* harbors three genes encoding members of the LDT family. We report a functional characterization of each of these three enzymes and the consequences of *ldt* gene deletions on antibiotic susceptibility, envelope stability, and *in vitro* formation of biofilms by *P. aeruginosa*.

## RESULTS

**The peptidoglycan of *P. aeruginosa* and *E. coli* have similar structures.** Our first objective was to determine the structure of peptidoglycan isolated from a stationary-phase culture of *P. aeruginosa* strain PA14. Peptidoglycan was extracted and digested with muramidases, and the resulting muropeptides were separated by reverse-phase high-performance liquid chromatography (rpHPLC) (Fig. 2A). Mass spectrometry (MS) analysis (Fig. 2B; see Table S1 in the supplemental material) was performed on the material that was collected in each of the individually collected peaks (Fig. 2, peaks 1 to 7). The most abundant stem peptide was a tetrapeptide (L-Ala–D-iGlu–DAP–D-Ala [where iGlu is isoglutamic acid]),

**TABLE 1** Muropeptide composition of peptidoglycan extracted from derivatives of *E. coli* Δ6*ldt* expressing *P. aeruginosa* L,D-transpeptidase genes

| Muropeptide (cross-link) | Calculated mass[a] | Observed mass[a] of muropeptides for strains with indicated deletion | | | |
|---|---|---|---|---|---|
| | | None | $ldt_{Pae1}$ | $ldt_{Pae2}$ | $ldt_{Pae3}$ |
| Monomers | | | | | |
| Tri | 870.371 | 870.372 | 870.371 | 870.370 | 870.371 |
| Tetra | 941.408 | 941.408 | 941.409 | 941.408 | 941.409 |
| Tri→KR | 1,154.567 | ND | ND | 1,154.567 | ND |
| | | | | | |
| Dimers | | | | | |
| Tri→Tri (3→3) | 1,722.731 | ND | 1,722.730 | ND | ND |
| Tetra→Tri (4→3) | 1,793.768 | ND | 1,793.765 | 1,793.765 | ND |
| Tri→Tetra (3→3) | 1,793.768 | ND | 1,793.765 | ND | ND |
| Tetra→Tetra (4→3) | 1,864.805 | 1,864.805 | 1,864.800 | 1,864.803 | 1,864.800 |

[a]Monoisotopic mass. ND, not detected. Data are representative of three biological repeats.

both in monomers and in the acceptor stem of dimers. The pentapeptide stem (L-Ala–D-iGlu–DAP–D-Ala–D-Ala) was not detected, indicating that the terminal D-alanines of stem pentapeptides that did not participate in peptidoglycan cross-linking were effectively trimmed off by D,D-carboxypeptidases. Cleavage of DAP–D-Ala amide bonds by L,D-carboxypeptidases or by endopeptidases generated the tripeptide stem L-Ala–D-iGlu–DAP. Most of the dimers (87%) contained 4→3 cross-links made by the D,D-transpeptidase activity of PBPs. The remaining dimers (13%) contained 3→3 cross-links formed by L,D-transpeptidases. Detection of the tripeptide L-Ala–D-iGlu–DAP linked to a Lys-Arg dipeptide (Tri→KR) revealed the anchoring of the OprI lipoprotein to the peptidoglycan (Fig. 1, reaction B). OprI is a homolog (24.8% identity) of the Lpp Braun lipoprotein of *E. coli* (Fig. S1). The disaccharide moiety was composed exclusively of GlcNAc and reduced MurNAc. Deacylated sugars, as previously reported (39, 40), were not detected, indicating that the corresponding muropeptides were present in insufficient amounts to be identified in our study design. Together, these results showed that all muropeptides detected in *E. coli* (23) are present in *P. aeruginosa* PA14. Conversely, no additional structure was detected in *P. aeruginosa*. Quantitatively, the muropeptide compositions were also very similar, indicating that the peptidoglycan structure is conserved in *P. aeruginosa* and *E. coli*.

**Heterospecific expression of *P. aeruginosa* *ldt* genes in *E. coli* reveals the function of the corresponding LDTs.** Our next objective was to identify the LDTs catalyzing the formation of 3→3 cross-linked muropeptides 4 and 6 (Fig. 2) and the anchoring of OprI to the peptidoglycan (Fig. 1, reactions A and B, respectively). Amino acid sequence comparisons using BLASTP as the software and LDTs from *E. coli* as the queries identified three proteins comprising an YkuD L,D-transpeptidase domain (protein family domain PF03734) with a conserved (S/T)XGCh(R/N) catalytic domain, in which C is the Cys nucleophile, X is any residue, and h is a hydrophobic residue (Fig. S2). The function of these enzymes, designated $Ldt_{Pae1}$ (PA14_54810), $Ldt_{Pae2}$ (PA14_27180), and $Ldt_{Pae3}$ (PA14_15840), was investigated by expressing the corresponding genes in a derivative of *E. coli* BW25113 obtained by deletion of the complete set of the six *E. coli* *ldt* genes (26), here designated *E. coli* Δ6*ldt*. The structure of peptidoglycan from this strain and derivatives independently producing each of the three LDTs from *P. aeruginosa* was determined based on purification of muropeptides by rpHPLC and determination of their structure by MS.

The peptidoglycan of *E. coli* Δ6*ldt* used for heterospecific *ldt* gene expression contained two main muropeptides, a tetrapeptide monomer and a 4→3 cross-linked Tetra→Tetra dimer formed by PBPs. Expression of *P. aeruginosa* $ldt_{Pae1}$ led to the formation of two additional 3→3 cross-linked dimers containing a tripeptide donor stem and a tripeptide or a tetrapeptide acceptor stem (Tri→Tri and Tri→Tetra dimers) (Fig. 2B, muropeptides 4 and 6; Table 1). These results indicate that $Ldt_{Pae1}$ is functional in *E. coli* and acts as a peptidoglycan cross-linking enzyme in this host.

Heterospecific expression of genes encoding $Ldt_{Pae2}$ in *E. coli* Δ6*ldt* led to the formation of an additional disaccharide-tripeptide monomer substituted by the dipeptide

**TABLE 2** Muropeptide composition of the peptidoglycan of derivatives of strain PA14 harboring various deletions

| Muropeptide (cross-link) | Calculated mass[a] | Observed mass[a] of muropeptides for strains with indicated deletion(s) | | | | | |
|---|---|---|---|---|---|---|---|
| | | None | $\Delta ldt_{Pae1}$ | $\Delta ldt_{Pae2}$ | $\Delta ldt_{Pae3}$ | $\Delta ldt_{Pae1}$ $\Delta ldt_{Pae2}$ $\Delta ldt_{Pae3}$ | $\Delta oprI$ |
| Monomers | | | | | | | |
| Tri | 870.371 | 870.370 | 870.369 | 870.371 | 870.370 | 870.374 | 870.364 |
| Tetra | 941.408 | 941.407 | 941.406 | 941.406 | 941.408 | 941.412 | 941.405 |
| Tri→KR | 1,154.567 | 1,154.566 | 1,154.565 | ND | 1,154.565 | ND | ND |
| Dimers | | | | | | | |
| Tri-Tri (3→3) | 1,722.731 | 1,722.731 | ND | ND | ND | ND | ND |
| Tetra-Tri (4→3) | 1,793.768 | 1,793.766 | 1,793.764 | ND | 1,793.767 | ND | 1,793.761 |
| Tri-Tetra (3→3) | 1,793.768 | 1,793.764 | ND | 1,793.764 | 1,793.766 | ND | 1,793.763 |
| Tetra-Tetra (4→3) | 1,864.805 | 1,864.802 | 1,864.801 | 1,864.799 | 1,864.804 | 1,864.818 | 1,864.800 |

[a]Monoisotopic mass. ND, not detected. Data are representative of three biological repeats except for strain $\Delta oprI$ (two repeats).

Lys-Arg (KR) (Fig. 2B, muropeptide 3; Table 1). This observation indicates that Ldt$_{Pae2}$ catalyzes the anchoring of Lpp despite the sequence divergence between the sequence of the *E. coli* (Lpp) and *P. aeruginosa* (OprI) lipoproteins (24.8% identity) (Fig. S1).

Heterospecific production of Ldt$_{Pae3}$ in *E. coli* Δ6*ldt* did not lead to any modification of the muropeptide profile. Thus, Ldt$_{Pae3}$ did not catalyze formation of 3→3 cross-links or the anchoring of the Braun lipoprotein in *E. coli*.

**Deletion of *P. aeruginosa ldt* genes confirms the function of Ldt$_{Pae1}$, Ldt$_{Pae2}$, and Ldt$_{Pae3}$ inferred from heterospecific gene expression.** rpHPLC chromatography and MS analyses revealed that deletion of *ldt$_{Pae1}$* abolished the formation of 3→3 cross-linked dimers, indicating that Ldt$_{Pae1}$ is the only peptidoglycan cross-linking L,D-transpeptidase produced by *P. aeruginosa* PA14 (Table 2). Deletion of *ldt$_{Pae2}$* alone abolished OprI anchoring, indicating that Ldt$_{Pae2}$ is the only enzyme responsible for the anchoring of that lipoprotein. Unlike with the parental strain, deletion of *ldt$_{Pae3}$* did not result in any modification of the muropeptides. Deletion of the three *ldt* genes abolished both the formation of 3→3 cross-links and the anchoring of the Lpp lipoprotein. Deletion of the *oprI* gene prevented formation of muropeptide 3 (Tri→KR; calculated monoisotopic mass [$M_{cal}$] = 1,154.567), confirming that this muropeptide originates exclusively from the anchoring of OprI to peptidoglycan. Together, these results (Table 2) indicate that formation of 3→3 cross-linked dimers and lipoprotein anchoring are mediated by Ldt$_{Pae1}$ and Ldt$_{Pae2}$, respectively, whereas Ldt$_{Pae3}$ catalyzes neither reaction, in agreement with the analysis of heterospecific expression of *P. aeruginosa ldt* genes in *E. coli* Δ6*ldt* (see above) (Table 1).

**Purified Ldt$_{Pae1}$ catalyzes the formation of 3→3 cross-linked dimers.** In order to validate the catalytic activity of Ldt$_{Pae1}$, we produced a soluble fragment of Ldt$_{Pae1}$ lacking the membrane anchor (residues 1 to 32) in *E. coli* and purified by metal affinity and size exclusion chromatography. Consistent with our previous results, Ldt$_{Pae1}$ was functional in the formation of 3→3 cross-linked dimers (Fig. 3) using tetrapeptide-containing peptidoglycan fragments as the substrates (Fig. 2B, muropeptide 2). The purified protein also catalyzed the exchanges of D-Ala$^4$ of these substrates by D-Met (Fig. 3). Ldt$_{Pae1}$ formed covalent links with β-lactams representative of the cephem (ceftriaxone) and carbapenem (meropenem) classes but not with the penam ampicillin. This β-lactam specificity was previously observed for LDTs from various bacteria (9–13, 41). We could not similarly investigate the catalytic activity of Ldt$_{Pae2}$ and Ldt$_{Pae3}$, since expression of fragments of the *ldt$_{Pae2}$* and *ldt$_{Pae3}$* genes under the conditions reported for *ldt$_{Pae1}$* did not afford soluble proteins.

**Ldt$_{Pae1}$ is unable to bypass β-lactam-inactivated PBPs.** We have previously shown that the D,D-transpeptidase activity of all PBPs can be replaced by the L,D-transpeptidase activity of one of the six LDTs of *E. coli*, namely, YcbB (7). This results in broad-spectrum β-lactam resistance, because YcbB is not effectively inactivated by β-lactams belonging to the penicillin and cephalosporin classes. In the presence of these drugs, all (>95%) of the peptidoglycan cross-links are of the 3→3 type, indicating that the PBPs do not contribute to peptidoglycan polymerization under such conditions. The bypass resistance mechanism

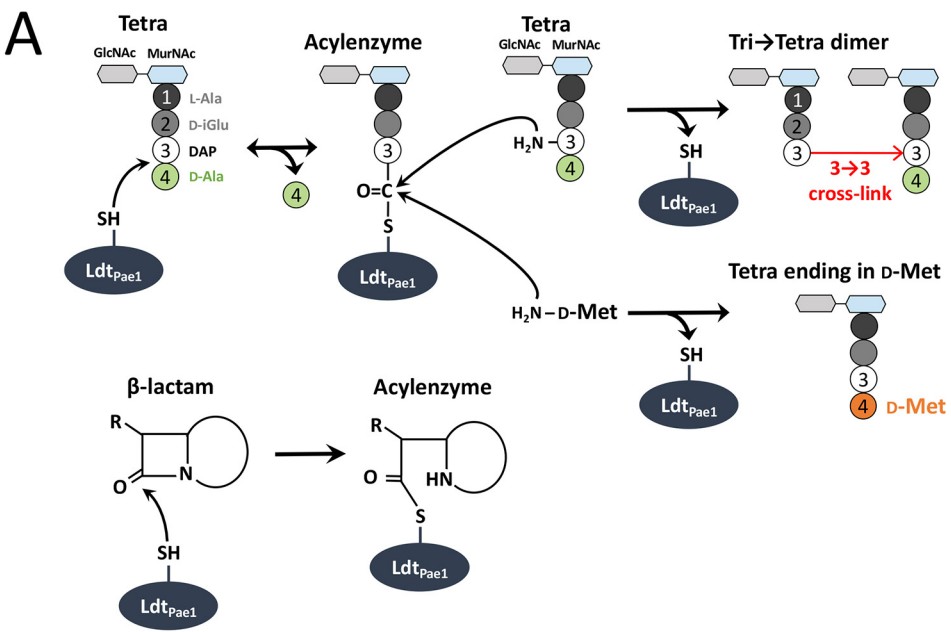

**FIG 3** Functional characterization of purified Ldt$_{Pae1}$. (A) Schematic representation of the reactions catalyzed by Ldt$_{Pae1}$. This enzyme forms an acyl-enzyme with a tetrapeptide donor stem that subsequently reacts with a tetrapeptide acyl acceptor or D-Met, resulting in the formation of a 3→3 cross-linked dimer or of a tetrapeptide ending in D-Met, respectively. Ldt$_{Pae1}$ also uses $\beta$-lactams as suicide substrates to form acyl-enzymes. (B) Identification of the reaction products by low-resolution MS. The indicated masses are monoisotopic masses for peptidoglycan fragments and average masses for acyl-enzymes. Tetra, reduced disaccharide-tetrapeptide; Lactoyl-Tetra, tetrapeptide linked to the D-lactoyl moiety of MurNAc following cleavage of the ether bond internal to MurNAc; D-Met, D-methionine; None, no reaction product obtained by incubation of purified Ldt$_{Pae1}$ with ampicillin. The calculated mass of the native protein is 36,303 Da, corresponding to residues 32 to 347 fused to the MGSSHHHHHHSSG His tag. The N-terminal methionine was not present in purified Ldt$_{Pae1}$.

also requires overproduction of the (p)ppGpp alarmone to prevent the bacterial killing triggered by inactivation of the transpeptidase domains of the PBPs (7, 42). This bactericidal activity of $\beta$-lactams is thought to result from the uncoupling of the transglycosylation and transpeptidation reactions, thereby leading to the accumulation of un-cross-linked glycan chains in the periplasm and the activation of a futile cycle of glycan chain polymerization and hydrolysis (42). Experimentally, overproduction of (p)ppGpp is achieved in an *E. coli* BW25113 derivative by the introduction of plasmid pKT8(*relA'*) encoding an unregulated (p)ppGpp synthase (RelA') and by the deletion of the chromosomally located *relA* gene encoding the wild-type (p)ppGpp synthase (7). Although L,D-transpeptidase YnhG from *E. coli* also catalyzes formation of 3→3 cross-links, this enzyme is unable to confer $\beta$-lactam resistance in the host overproducing (p)ppGpp (7). We therefore investigated whether Ldt$_{Pae1}$, which forms 3→3 cross-links, was able to bypass PBPs and confer $\beta$-lactam resistance in *E. coli*. To address this question, we introduced plasmid pIHB1, which

**TABLE 3** Expression of ʟ,ᴅ-transpeptidase-mediated resistance in (p)ppGpp-producing *E. coli*[a]

| Plasmid | Inhibition zone size (mm) around disks containing indicated β-lactam | |
| --- | --- | --- |
| | Ceftriaxone (30 μg) | Ampicillin (10 μg) |
| pHV6 (vector) | 37 | 23 |
| pHV6 (*ycbB*) | 12 | ≤6 |
| pHV6 (*ynhG*) | 38 | 21 |
| pHV6 (*ldt*$_{Pae1}$) | 39 | 20 |

[a]The *E. coli* host harbored a chromosomal deletion of *relA* and plasmid pKT8 encoding RelA′. Data are median values of three biological repeats.

carries a copy of the *ldt*$_{Pae1}$ gene under the control of an isopropyl-β-ᴅ-thiogalactopyrano-side (IPTG)-inducible promoter, in the derivative of *E. coli* BW25113 that overproduces the (p)ppGpp alarmone. Plasmids coding for inducible production of YcbB and YnhG were used as positive and negative controls, respectively. Production of *P. aeruginosa* Ldt$_{Pae1}$ did not mediate β-lactam resistance in this host, as previously found for *E. coli* YnhG (Table 3). Thus, YnhG and Ldt$_{Pae1}$ were unable to cross-link a functional peptidoglycan under conditions in which the ᴅ,ᴅ-transpeptidase activity of PBPs was inactivated by β-lactams, even though both proteins are functional LDTs for synthesis of 3→3 peptidoglycan cross-links *in vitro* and in *E. coli*. Similar to PBPs involved in peptidoglycan cross-linking, YcbB is a high-molecular-weight protein composed of 615 residues that contains several domains not present in other LDTs, including LDT$_{Pae1}$ and YnhG (Fig. S4). These domains are likely to be essential for the bypass of PBPs by enabling YcbB to recruit accessory proteins required to carry out peptidoglycan cross-linking during the entire cell cycle. These accessory proteins may include the bifunctional transpeptidase-glycosyltransferase class A PBP1b, which displays affinity for YcbB (26), and scaffolding proteins controlling the sites of peptidoglycan expansion during the cell cycle (7, 42).

**The activity of β-lactams is not affected by deletion of *ldt* genes.** The activity of 31 antibiotics, including 20 β-lactams, was tested by the disk diffusion assay against *P. aeruginosa* PA14 and isogenic derivatives obtained by deletion of ʟ,ᴅ-transpeptidase genes (Table S3). Deletion of *ldt*$_{Pae1}$, *ldt*$_{Pae2}$ plus *ldt*$_{Pae3}$, or of the three ʟ,ᴅ-transpeptidase genes did not affect the susceptibility of *P. aeruginosa* to any of the tested antibiotics. Thus, the ʟ,ᴅ-transpeptidases did not contribute to growth of PA14 in the presence of subinhibitory concentrations of β-lactams or to the efficacy of the permeability barrier mediated by the outer membrane.

**Deletion of *ldt*$_{Pae2}$ destabilizes the *P. aeruginosa* envelope.** EDTA is often used to probe the stability of the outer membrane of *E. coli*, since this compound prevents the stabilization of the outer leaflet of the outer membrane by $Mg^{2+}$ cations (30, 42). Incubation of *P. aeruginosa* PA14 derivatives lacking *oprl* or *ldt*$_{Pae2}$ resulted in similar kinetics of bacterial killing by EDTA with ∼6-$\log_{10}$ decreases of the initial inoculum (Fig. 4). Deletion of *ldt*$_{Pae1}$ or of *ldt*$_{Pae3}$ had no impact on killing in comparison to that of wild-type PA14, with <2-$\log_{10}$ decreases of the initial inoculum. Thus, loss of Oprl or of the anchoring of this lipoprotein to the peptidoglycan destabilizes the cell envelope, leading to cell death in the presence of EDTA.

**Deletion of *ldt* genes impairs biofilm formation but does not affect motility.** Screening of a library of mutants obtained by random insertions of a transposon previously revealed that a member of the LDT family (YafK) is required for biofilm formation by an enteroaggregative strain of *Escherichia coli* (43). This prompted us to determine the impact of *ldt* deletions on biofilm formation by *P. aeruginosa*.

Our first objective was to evaluate the contribution of LDTs to peptidoglycan synthesis in biofilm (Table S2; Fig. 5). The proportion of 3→3 cross-linked dimers among all dimers was 11.3% in peptidoglycan extracted from biofilms, whereas 3→3 cross-links were not detected for exponential growth of PA14 in planktonic form (Fig. 5B). The relative proportion of dimers among all muropeptides was higher in biofilm than in planktonic cultures (44.2% versus 33.9%) (Fig. 5A). These results indicate that growth

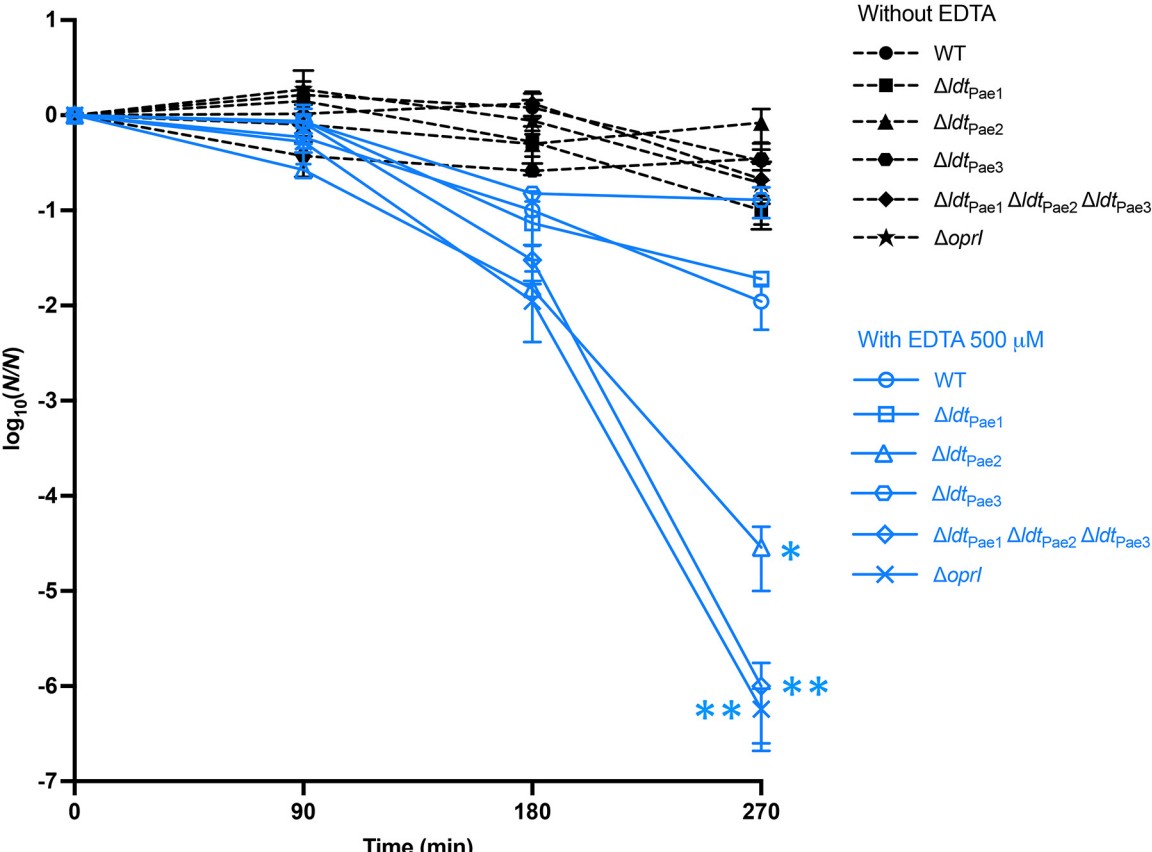

**FIG 4** Killing of *P. aeruginosa* by EDTA. Bacteria from overnight cultures were resuspended in Tris-HCl, pH 8.0, and incubated with 500 $\mu$M EDTA (open blue symbols and solid lines) or without EDTA (filled black symbols and dotted lines). Bacteria were enumerated at 0, 90, 180, and 270 min. The numbers of CFU were normalized to the starting inoculum. Data are the mean $\pm$ standard error of the mean from three independent biological repeats. * and **, $P < 0.05$ and $P < 0.01$, respectively (Brown-Forsythe and Welch ANOVA [wild-type with EDTA versus mutants with EDTA at 270 min]).

in biofilm favors peptidoglycan cross-linking by $Ldt_{Pae1}$ and that this growth condition was associated with a highly cross-linked peptidoglycan. Growth in biofilm also favored the anchoring of OprI to peptidoglycan, indicating that $Ldt_{Pae2}$ was active under this growth condition (Fig. 5C).

Our second objective was to determine whether the activity of the LDTs modulate biofilm formation. The deletion of $ldt_{Pae1}$, $ldt_{Pae2}$, $ldt_{Pae3}$, or *oprl* resulted in statistically significant but limited decreases in the capacity of *P. aeruginosa* PA14 to form biofilm in the 96-well plate assay (Fig. 5D). The combine deletions of all three L,D-transpeptidases did not further decrease biofilm formation. Additionally, we also demonstrated that deletions of *ldt* genes or of *oprl* had no impact on the swimming motility of PA14, a biofilm-related function (Table S4), indicating that the cell envelope of the mutants was fully compatible with the assembly of a functional flagellum.

**Comparison of LDTs catalyzing various reactions.** A peptidoglycan cross-linking activity was assigned to $Ldt_{Pae1}$, a lipoprotein-anchoring activity was assigned to $Ldt_{Pae2}$, and neither activity was assigned to $Ldt_{Pae3}$ (see above) (Tables 1 and 2). The resulting extension in the number of functionally characterized LDTs prompted us to explore the phylogenetic relationships between members of this protein family (Table S5; Fig. S2 and S3). Closely related species were found to produce the same number of closely related LDTs, which share sequence identity over the entire sequences (e.g., *E. coli* and *Salmonella enterica* [6 paralogues], *E. faecalis* and *E. faecium* [1 paralogue], and *Mycobacterium tuberculosis* and *Mycobacterium abscessus* [6 paralogues]). More distantly related species, such as *E. coli* and *P. aeruginosa* or *E. faecium* and *S. aureus*

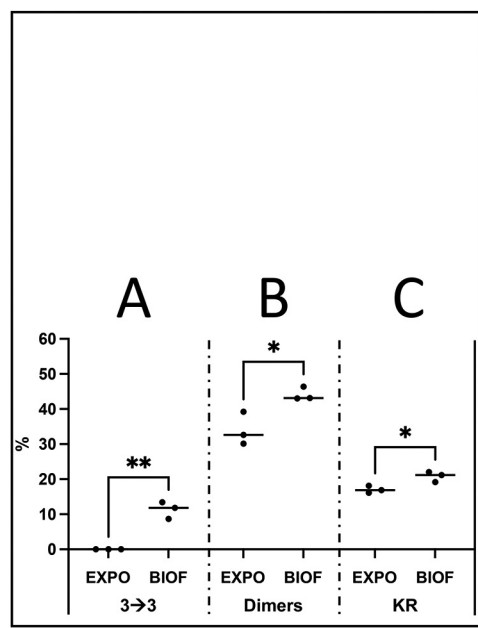
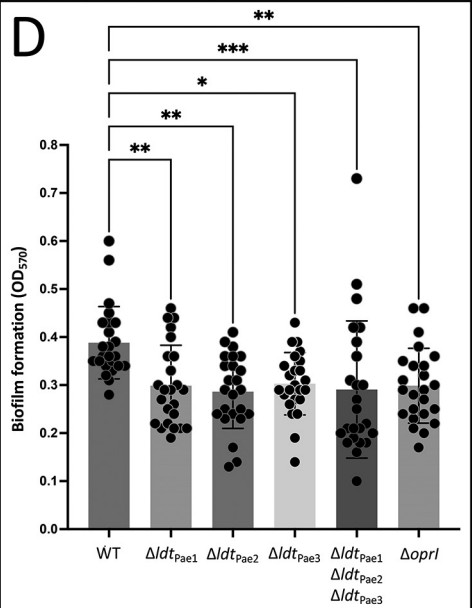

**FIG 5** Peptidoglycan structure of *P. aeruginosa* grown in biofilm and impact of *ldt* gene deletion on biofilm formation. (A) Proportion of dimers with 3→3 cross-links among all dimers (cumulative area of peaks 4 and 6 divided by the cumulative area of peaks 4 to 7 in rpHPLC chromatograms; λ = 205 nm). (B) Proportion of dimers among all muropeptides except peak 3 (cumulative area of peaks 4 to 7 divided by cumulative area of peak 1 to 7, except peak 3). (C) Proportion of muropeptides containing the Lys-Arg fragment of Oprl among all muropeptides (cumulative area of peak 3 divided by cumulative area of peaks 1 to 7). Filled circles indicate the values obtained for biological repeats ($n = 3$). Horizontal lines indicate the mean percent values. * and **, $P < 0.05$ and $P < 0.01$, respectively; unpaired *t* test. BIOF, PA14 grown for 96 h in biofilm; EXPO, exponentially growing PA14 in planktonic form ($OD_{600}$, 0.8). (D) Crystal violet staining of biofilms formed by wild-type *P. aeruginosa* PA14 and derivatives obtained by deletion of *ldt* genes and *oprl*. Circles indicate values obtained for biological repeats ($n = 24$). *, **, and ***, $P < 0.05$, $P < 0.01$, and $P < 0.001$, respectively (Brown-Forsythe and Welch ANOVA test).

(which does not harbor any LDT), produce various numbers of LDTs (0 to 6), and the domain composition is not conserved (Fig. S4).

We then evaluated whether LDTs catalyzing similar reactions could be identified based on the presence of specific sequence signatures. Winkle et al. identified a polymorphism in the conserved catalytic motif of *E. coli* LDTs (SXGChR versus SXGChA) and proposed that the latter motif was specific for enzymes hydrolyzing the amide bond connecting tripeptide stems to lipoproteins (YafK, also referred to as DpaA, in *E. coli*) (32, 33). Ldt$_{Pae3}$ might be involved in the releasing of anchored Oprl in *P. aeruginosa*, since this L,D-transpeptidase contains an SXGCh<u>A</u> motif and is closely related to YafK (Table S5; Fig. S2 and S3). Ldt$_{Pae2}$ (Oprl anchoring) belonged to a lineage that included both *E. coli* Lpp anchoring (ErfK, YbiS, and YcfS) and peptidoglycan cross-linking (YnhG) enzymes. For Ldt$_{Pae1}$ (formation of 3→3 cross-links), the highest levels of identity were observed for YciB from *Bacillus subtilis* (35.6%; unknown function), Ldt$_{Mab3}$ from *Mycobacterium abscessus* (32.9%; unknown function), and YcfS of *Campylobacter jejuni* and *E. coli* (31.2% and 30.1%, unknown function and Lpp anchoring, respectively). Together, these results imply that LDTs with lipoprotein anchoring and 3→3 cross-linking activities cannot be assigned to two divergent evolutionary lineages, because they are scattered in the various branches of the phylogram depicted in Fig. S3.

**Ldt$_{Pae1}$ comprises two YkuD domains.** Sequence comparison indicated that Ldt$_{Pae1}$, which catalyzes the formation of 3→3 peptidoglycan cross-links, comprises an N-terminal bona fide LDT catalytic domain with a conserved catalytic motif (SHGCIR), followed by a related domain that lacks the catalytic Cys residue (QLGKIR) (these domains are designated Ldt$_{Pae1}$ and Ldt$_{Pae11}$, respectively, in Table S5 and Fig. S2 and S3 and YkuD1 and YkuD2 in Fig. S4). These two domains are clustered in the same lineage, suggesting a duplication. This domain architecture has not been detected in any other LDT

sequence. The role of the catalytically deficient domain Ldt$_{Pae11}$ remains to be determined. It is tempting to speculate that it could be involved in peptidoglycan binding, a function mediated by unrelated domains in other LDTs (44, 45).

## DISCUSSION

Active-site cysteine L,D-transpeptidases comprising a conserved YkuD domain catalyze various reactions (Fig. 1; see also the introduction) (46). Here, we show that *P. aeruginosa* may produce representatives of each of the three catalytic functions associated with the conserved YkuD catalytic domain, namely, the formation of 3→3 peptidoglycan cross-links (Ldt$_{Pae1}$), the anchoring of lipoprotein OprI to peptidoglycan (Ldt$_{Pae2}$), and the hydrolysis of the resulting tripeptide→OprI amide bond (Ldt$_{Pae3}$). The activity of these enzymes was investigated by *in vitro* assays for purified Ldt$_{Pae1}$ (Fig. 3), deletion of *ldt* genes from the chromosome of *P. aeruginosa* PA14 (Table 2), and heterologous expression of *ldt* genes in *E. coli* (Table 1). The functionality of Ldt$_{Pae1}$ and Ldt$_{Pae2}$ in the heterologous host might be accounted for by the conserved structures of both the peptidoglycan subunit of *P. aeruginosa* and *E. coli* (Fig. 2) and the Arg-Lys C terminus of lipoproteins OprI and Lpp (see Fig. S1 in the supplemental material). The function of Ldt$_{Pae3}$ was tentatively assigned to the release of OprI based on both the absence of detectable amide bond-forming activity (formation of 3→3 and tripeptide→OprI bonds) and the close similarity between Ldt$_{Pae3}$ and *E. coli* hydrolase YafK.

Phylogenetic analysis revealed that the number of LDTs and their domain composition are not conserved except in closely related bacteria belonging to the same genus (Table S5 and Fig. S2 to S4). LDTs with lipoprotein anchoring and 3→3 cross-linking activities cannot be assigned to two distinct evolutionary lineages. Thus, although *ldt* genes appear to belong to bacterial core genomes, they are not stably inherited in distantly related lineages. In contrast, comparison of high-molecular-weight PBPs involved in peptidoglycan cross-linking clearly identifies orthologues in distantly related lineages (47).

The phenotypic impact of deletions of *ldt* genes and of *oprI* has been explored using various assays. The deletion of these genes did not result in hypersusceptibility to antibiotics or impaired motility (Tables S3 and S4, respectively). These observations indicate that the permeability barrier of the outer membrane and the functionality of the flagellum were preserved in the mutants. Loss of OprI or of the anchoring of this lipoprotein to peptidoglycan led to a destabilization of the cell envelope, as revealed by the bactericidal effect of EDTA (Fig. 4). In contrast, formation of 3→3 cross-links by Ldt$_{Pae1}$ was dispensable for survival in the presence of EDTA. Similar results were previously obtained for mutants of *E. coli* deficient in the production of Lpp or in the anchoring of this lipoprotein to peptidoglycan (30).

Formation of 3→3 cross-links remained undetected for exponential growth of *P. aeruginosa* PA14 in planktonic form (Fig. 5B). The proportion of 3→3 cross-links was significantly higher (11.3%) for growth in biofilm ($P < 0.01$). Deletion of *ldt*$_{Pae1}$ had a modest but significant ($P < 0.05$) impact on the ability of *P. aeruginosa* PA14 to form biofilms (Fig. 5D). These results indicate that 3→3 cross-linking of glycan strands participates in the adaptation of peptidoglycan metabolism to conditions that prevail in biofilm. Interestingly, muropeptides containing 3→3 cross-links were found to be more abundant in *P. aeruginosa* epidemic strains than in reference strains PAO1 and PA14 (40). In addition, deletion of ldt$_{Pae2}$ or *oprI* impaired biofilm formation and sensitized mutants to EDTA. Thus, L,D-transpeptidases might be actionable targets to fight against bacteria in biofilm. Considering the impact of *ldt* gene deletions on the bactericidal activity of EDTA (Fig. 4), it would be relevant to determine whether inhibition of LDTs of Gram-negative bacteria could act in synergy with drug-EDTA associations for the eradication of catheter-associated biofilms based on antibiotic lock therapy or in treatment of other biofilm-related infections (48–50).

## MATERIALS AND METHODS

**Bacterial strains and plasmids.** The characteristics and origin of plasmids and bacterial strains are given in Tables S6 and S7, respectively, in the supplemental material.

**Cultures for peptidoglycan extraction.** For the analyses of the muropeptide composition of the peptidoglycan from *P. aeruginosa* PA14 and derivatives obtained by deletion of *ldt* genes and *oprl*, bacteria were grown overnight to stationary phase in 200 mL lysogeny (Miller) broth (LB) at 37°C. For quantitative analyses of the peptidoglycan of *P. aeruginosa* PA14 in the exponential phase of growth, bacteria were grown in 1 L of LB broth and collected at an optical density at 600 nm ($OD_{600}$) of 0.8. For quantitative analyses of the peptidoglycan of *P. aeruginosa* PA14 grown in biofilm, continuous-flow biofilm microfermentors containing a removable glass spatula were used as described previously (51). Biofilm microfermentors were inoculated by placing the spatula in a culture solution adjusted to an $OD_{600}$ of 1.0 (ca. $5.0 \times 10^8$ bacteria/mL) for 5 min. The spatula was placed into the microfermentor, and biofilm culture was performed at 37°C in Miller LB broth. The flow rate was adjusted so that the total time for renewal of microfermentor medium was shorter than the bacterial generation time, thus minimizing planktonic growth by constant dilution of nonbiofilm bacteria. Biofilms were allowed to grow on the glass spatula for 72 h, after which the microfermentor was vortexed for 1 min to resuspend the bacterial population. The resulting bacterial suspension (50 mL) was centrifuged for 15 min at $7,000 \times g$ (4°C), and peptidoglycan was extracted by the hot SDS procedure. Peptidoglycan analyses were performed for a minimum of three biological repeats.

**Analysis of peptidoglycan structure.** Sacculi were extracted by the hot SDS procedure and treated with pronase and trypsin (7, 52). Muropeptides were solubilized by digestion with muramidases, reduced with $NaBH_4$, and purified by rpHPLC. The mass of muropeptides was determined on a Bruker Daltonics maXis high-resolution MS (Bremen Germany) operating in the positive mode, as previously described (53, 54).

**Heterospecific expression of *P. aeruginosa* *ldt* genes in *E. coli* BW25113 Δ6*ldt*.** Genes $ldt_{Pae1}$, $ldt_{Pae2}$, and $ldt_{Pae3}$ were cloned into the vector pHV6 under the control of the *trc* promoter by Gibson assembly. Genes encoding LDTs were induced at an $OD_{600}$ of 0.2 with 100 $\mu$M IPTG in LB broth. The incubation was continued for 18 h at 37°C.

**Purification of $Ldt_{Pae1}$.** A fragment of the $ldt_{Pae1}$ gene encoding residues 33 to 347 of the L,D-transpeptidase was cloned into the vector pET-TEV, generating a translational fusion with a C-terminal 6×His tag. The protein was produced in *E. coli* BL21 and purified from a clarified lysate by metal affinity and size exclusion chromatography as previously performed for L,D-transpeptidase YcbB from *E. coli* (7). The protein concentration was determined by the Bio-Rad assay, using bovine serum albumin as the standard.

**Peptidoglycan transpeptidase activity of purified $Ldt_{Pae1}$.** Reduced disaccharide-tetrapeptide and lactoyl-tetrapeptide were extracted from the peptidoglycan of *E. coli* BW25113 Δ6*ldt* as previously described (55). Purified $Ldt_{Pae1}$ (5 $\mu$M) was incubated with peptidoglycan fragments (50 $\mu$M) in 25 mM Tris-HCl (pH 8.0) at 20°C. LC-MS was performed with a Nucleoshell RP 18 column (5 $\mu$m; 50 mm by 2 mm) coupled to a low-resolution LCQ Advantage mass spectrometer (ThermoElectron). The reactions were also carried out in the presence of additional D-Met (1 mM).

**MS analyses of $Ldt_{Pae1}$ acylation by $\beta$-lactams.** The formation of drug-enzyme adducts was tested by incubating $Ldt_{Pae1}$ (10 $\mu$M) with $\beta$-lactams (100 $\mu$M) at 20°C in 5 mM Tris-HCl (pH 8.0) (9). Five microliters of acetonitrile and 1 $\mu$L of 1% formic acid were added, and the reaction mixture was directly injected into the MS. Spectra were acquired in the positive mode on a Bruker Daltonics maXis high-resolution MS (Bremen, Germany) operating in the positive mode.

**Deletion of genes $ldt_{Pae1}$, $ldt_{Pae2}$, $ldt_{Pae3}$, and *oprl* of *P. aeruginosa* PA14.** The two-step allelic exchange procedure was used for deleting *ldt* genes and *oprl* (56). Briefly, sequences flanking the genes were cloned into the vector pEX18. The plasmids were transferred by conjugation from *E. coli* S17 $\lambda$pir to *P. aeruginosa* PA14. Transconjugants were selected on LB agar plates containing 15 $\mu$g/mL triclosan and 75 $\mu$g/mL tetracycline. Colonies were subcultured in salt-free LB containing 5% sucrose. The genome of the mutants was sequenced by the Illumina approach (Novogene, Cambridge, UK).

**Antimicrobial susceptibility testing.** Antibiograms were performed in triplicate by the disk diffusion assay in cation-adjusted Mueller-Hinton agar plates according to the guidelines of the European Committee on Antimicrobial Susceptibility Testing (EUCAST) (57).

**Bactericidal activity of EDTA.** Bacteria were grown overnight to the stationary phase in LB (Miller) broth at 37°C, washed twice in 50 mM Tris-HCl buffer (pH 8.0), and resuspended in the same buffer to a final $OD_{600}$ of 0.9. EDTA was added (final concentration, 500 $\mu$M), bacteria were incubated in 2-mL Eppendorf tubes at 37°C, and samples were withdrawn at 90 min, 180 min, and 270 min for viable cell counts on agar plates. Data were obtained in three biological repeats.

**Plate-based assay for swimming motility.** An isolated colony from an overnight agar plate was inoculated onto a soft agar plate containing tryptone broth (10 g/L), NaCl (5 g/L), and agar (0.3%, wt/vol) (58). The plates were incubated for 18 h at 30°C, and the swimming diameter was recorded. Data were obtained in three biological repeats.

**Quantification of biofilm.** Formation of biofilm was determined by crystal violet staining in 96-well microtiter plates, as previously described (59). A microtiter plate was inoculated with 100 $\mu$L of overnight cultures at a dilution of 1:100 and incubated at 37°C for 24 h. The medium was removed, and the wells were washed twice with water. A 125-$\mu$L volume of a crystal violet solution (RAL Diagnostics) was added to each well, and the incubation was continued for 15 min. After washing three times with water, 125 $\mu$L of a mixture of ethanol and acetone (80/20, vol/vol) was added, and incubation was continued for 10 min. Absorbance was measured at 570 nm with a plate reader (Tecan Infinite 200Pro). Data were obtained in 24 biological repeats.

**Statistical analysis.** The data were analyzed using GraphPad Prism version 9 (La Jolla, CA, USA). An unpaired *t* test was performed to analyze the relative abundance of dimers and 3→3 cross-links in the peptidoglycan of *P. aeruginosa* PA14. A Brown-Forsythe and Welch analysis of variance (ANOVA) test was

performed to analyze the impact of *ldt* gene deletions on bacterial killing by EDTA and the quantification of biofilm formation. A *P* value of 0.05 was considered statistically significant in both types of tests.

**Sequence comparison and phylogeny.** Forty YkuD-related domains were retrieved from the genome of 12 bacterial species, including *E. coli* (*E. coli*_YcfS, *E. coli*_YcbB, *E. coli*_YnhG, *E. coli*_Ybis, *E. coli*_ErfK, and *E. coli*_YafK), *P. aeruginosa* (Ldt$_{Pae1}$ and its second YkuD domain named Ldt$_{Pae11}$, Ldt$_{Pae2}$, and Ldt$_{Pae3}$), *Salmonella enterica* serovar Typhimurium (Sty_YcfS, Sty_YcbB, Sty_YnhG, Sty_Ybis, Sty_ErfK, and Sty_YafK), *Campylobacter jejuni* (Cje_YcfS and Cje_Ybis), *Neisseria meningitidis* (Ldt$_{Nme}$), *Bacillus subtilis* (Bs_YkuD, Bs_YqjB, and Bs_YciB), *Clostridioides difficile* (Ldt$_{Cd1}$, Ldt$_{Cd2}$, and Ldt$_{Cd3}$), *Mycobacterium tuberculosis* (Ldt$_{Mt1}$ to Ldt$_{Mt6}$), *Mycobacterium abscessus* (Ldt$_{Mab1}$ to Ldt$_{Mab6}$), *Enterococcus faecium* (Ldt$_{fm}$), *Enterococcus faecalis* (Ldt$_{fs}$), and *Coxiella burnettii* (Ldt$_{Cbu2}$). The sequences of the domains were aligned using Clustal Omega (Fig. S2), providing the identity matrix reported in Table S5. The phylogram reported in Fig. S3 was obtained on iTOL (60).

## SUPPLEMENTAL MATERIAL

Supplemental material is available online only.
**SUPPLEMENTAL FILE 1**, PDF file, 1.2 MB.

## ACKNOWLEDGMENTS

High-resolution mass spectra were obtained at the Plateforme de Spectrométrie de Masse Bio-organique of the Muséum d'Histoire Naturel. We thank A. Filloux for the generous gift of *P. aeruginosa* mutants obtained by transposon insertion (full library available at http://pa14.mgh.harvard.edu/cgi-bin/pa14/mutantrequest.cgi).

This work was supported by the French National Research Agency (ANR) (grant no. ANR-19-CE15-0006-01, PeptidoAdapt, Program AAPG 2019 to M.A. and J.-M.G.). This work was also supported by the French government's Investissement d'Avenir Program, Laboratoire d'Excellence Integrative Biology of Emerging Infectious Diseases (grant no. ANR-10-LABX-62-IBEID). S.T.-R was supported by the ANR, project EvolTolAB (grant no. ANR-18-CE13-0010).

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
