## [Reviewer comments · Microbiology Spectrum]

Microbiology Spectrum

Characterization of *Pseudomonas aeruginosa* L,D-transpeptidases and evaluation of their role in peptidoglycan adaptation to biofilm growth

Inès Hugonneau-Beaufet, Jean-Philippe Barnier, Stanislas Thiriet-Rupert, Sylvie Létoffé, Jean-luc Mainardi, Jean-Marc GHIGO, Christophe Beloin, and Michel Arthur

Corresponding Author(s): Michel Arthur, INSERM

Review Timeline:

Submission Date:	December 20, 2022
Editorial Decision:	March 8, 2023
Revision Received:	April 16, 2023
Accepted:	April 18, 2023

Editor: Eric Cascales

Reviewer(s): Disclosure of reviewer identity is with reference to reviewer comments included in decision letter(s). The following individuals involved in review of your submission have agreed to reveal their identity: Erin Gloag (Reviewer #2)

Transaction Report:

DOI: <https://doi.org/10.1128/spectrum.05217-22>

March 8, 2023

Dr. Michel Arthur
INSERM
Paris
France

Re: Spectrum05217-22 (Characterization of *Pseudomonas aeruginosa* L,D-transpeptidases and evaluation of their role in peptidoglycan adaptation to biofilm growth)

Dear Michel,

Thank you for submitting your manuscript to Microbiology Spectrum. The review process has been now completed, and I again wish to apologize for the overly long delay. Your manuscript has been sent to two external reviewers. As you will see in their comments pasted below, the two reviewers have a very positive appraisal of your work. While reviewer #1 enjoyed reading your paper and suggests a few clarifications, reviewer #2 raised significant criticisms notably regarding data analyses and interpretation and requests more information on the rationale of some experiments. I therefore encourage you to carefully address the reviewer's comments and invite you to submit a revised version of your work.

Link Not Available

Sincerely,
Eric

Eric Cascales

Journals Department
Reviewer comments:

Reviewer #1 (Comments for the Author):

This is a very elegant study of the LDT homologue genes in *P. aeruginosa*. While the work is more comparative than groundbreaking, it makes a very valuable contribution to our understanding of the functions of the LDTs in an important pathogen. I enjoyed reading this paper, which has been carefully and clearly written. The experimental work is excellent and very nicely presented.

Minor comment:

As a non specialist, the section dealing with heterologous expression of LdtPae1 to bypass β -lactam-inhibited PBPs is difficult to follow and more detail is needed. Overproduction of the YcbB and (p)ppGpp in *E. coli* results in high-level β -lactam resistance due to bypass of inhibited PBPs. In this section of the paper, it is not clear why a LdtPae1 plasmid was introduced into an *E. coli* strain with plasmids encoding YcbB and YnhG (and constitutive RelA). Can the authors please elaborate on the rationale for this experimental set up.

Reviewer #2 (Comments for the Author):

Here authors describe the role of LDTs in *P. aeruginosa* peptidoglycan synthesis. Authors also identify that LdtPae2 is important for maintaining cell envelope integrity and LdtPae1 and LdtPae2 have increased activity during biofilm formation. This manuscript is well written and the results flow logically. My only main comment is to provide more detail regarding the analyses performed, how data is interpreted and the authors rationale for each results section, to improve clarity and understanding. My specific comments are below.

Major comments

1. Figure 1. Naïve question. Do the peaks in 1A correspond to each column in 1B? Perhaps explaining this analysis in more detail in L157 - 164 would be helpful.
2. L186. Can authors explain in more detail what analysis was performed for 'sequence comparisons'? Was this determined from the mass spec analysis?
3. L193 - 207. What analyses were performed to determine these results? Was the analysis presented in Fig 2 repeated here for the *E. coli* strain? Same comment for L211 - 223.
4. L235 - 236. Can authors expand on this? Were authors unable to make soluble fragments as they did for Ldt1?
5. L247 - 251. Can authors provide a bit more of a rationale here? Why did authors think that Ldt1 could replace YcbB? Is Ldt1 homologous to YcbB? What is YnhG? Why was it included here?
6. L259 - 260. Adjust the subheading, as authors do not present any motility data here.
7. Figure 4. In the legend describe how the data is plotted on the y-axis. Is this normalized to the starting inoculum? Is this CFU data?
8. L343 - 344. Why is this? Elaborate on this conclusion, and how it was drawn from the data.
9. L388 - 395. Do authors predict this is due to the sessile behaviour of bacteria in a biofilm? Do authors predict that stationary phase bacteria would have similar profiles?
10. Methods section. For each assay indicate the number of technical and biological replicates.

Minor comments

1. L33. LdtPae3?
2. L185. Indicate that 4 and 6 muropeptides were those identified in Fig 2 for clarity.
3. L187. What does 'PF03734' refer to?
4. L189 - 190. Can authors include the PA14 locus tag for each ldt?
5. Figure 3A is somewhat confusing. Are the reactions catalyzed by Ldt1 that indicated in red text, or where the navy circle is indicated? Expanding the figure legend to provide more detail would be helpful.
6. L231 - 234. Refer to Fig2B here.
7. L292. Specify that the 11.3% dimers were detected in biofilms, for clarity.
8. Figure 5D. * are mentioned in the figure legend, but there are none on the graph.

Staff Comments:

Preparing Revision Guidelines

Please return the manuscript within 60 days; if you cannot complete the modification within this time period, please contact me. If you do not wish to modify the manuscript and prefer to submit it to another journal, please notify me of your decision immediately so that the manuscript may be formally withdrawn from consideration by Microbiology Spectrum.

Responses to the reviewers

Reviewer #1:

Minor comment:

As a non specialist, the section dealing with heterologous expression of LdtPae1 to bypass β -lactam-inhibited PBPs is difficult to follow and more detail is needed. Overproduction of the YcbB and (p)ppGpp in *E. coli* results in high-level β -lactam resistance due to bypass of inhibited PBPs. In this section of the paper, it is not clear why a LdtPae1 plasmid was introduced into an *E. coli* strain with plasmids encoding YcbB and YnhG (and constitutive RelA). Can the authors please elaborate on the rationale for this experimental set up.

Answer: We have rephrased and expended this section to provide a more detailed background information and rationale for doing experiments (Revised manuscript lines 224 to 258).

Reviewer #2:

Major comments

1. Figure 1. Naïve question. Do the peaks in 1A correspond to each column in 1B? Perhaps explaining this analysis in more detail in L157 - 164 would be helpful.

Answer: The analysis of mucopeptides was based on their separation by *rp*HPLC, collect of individual peaks, and identification by mass spectrometry. The approach has been described in more details Lines 148 to 150 of the revised manuscript. We have also clearly indicated the correspondence between peaks and chemical structures in panels 2A and 2B of Fig. 1.

2. L186. Can authors explain in more detail what analysis was performed for 'sequence comparisons'? Was this determined from the mass spec analysis?

Answer: We have indicated that the analysis was based on amino acid sequence comparisons and briefly mentioned the procedure that was used (Lines 171 and 172 of the revised manuscript).

3. L193 - 207. What analyses were performed to determine these results? Was the analysis presented in Fig 2 repeated here for the *E. coli* strain? Same comment for L211 - 223.

Answer: The function of the LDTs was deduced from the structural analysis of the peptidoglycan by *rp*HPLC and mass spectrometry as indicated in the additional sentences appearing Lines 179 to 181 and Lines 198 and 199 of the revised manuscript.

4. L235 - 236. Can authors expand on this? Were authors unable to make soluble fragments as they did for Ldt1?

Answer: We have indicated lines 221 to 223 of the revised manuscript that the catalytic activity of Ldt_{Pae2} and Ldt_{Pae3} could not be analyzed since expression of fragments of the *ldt*_{Pae2} and *ldt*_{Pae3} genes in the conditions reported for *ldt*_{Pae1} did not afford soluble proteins

5. L247 - 251. Can authors provide a bit more of a rationale here? Why did authors think that Ldt1 could replace YcbB? Is Ldt1 homologous to YcbB? What is YnhG? Why was it included here?

Answer: This comment has been addressed in the response to reviewer 1.

6. L259 - 260. Adjust the subheading, as authors do not present any motility data here.

Answer: The subheading has been modified as requested (Line 259 of the revised manuscript).

7. Figure 4. In the legend describe how the data is plotted on the y-axis. Is this normalized to the starting inoculum? Is this CFU data?

Answer: The numbers of CFUs were normalized to the starting inoculum. The information was introduced in the legend to the figure of the revised version (Line 524).

8. L343 - 344. Why is this? Elaborate on this conclusion, and how it was drawn from the data.

Answer: We have indicated that LDTs with lipoprotein anchoring and 3→3 cross-linking activities cannot be assigned to two evolutionary divergent lineages because they are scattered in the various branches of the phylogram depicted in Supplementary Fig S3 (Lines 323 and 324).

9. L388 - 395. Do authors predict this is due to the sessile behavior of bacteria in a biofilm? Do authors predict that stationary phase bacteria would have similar profiles?

Answer: We currently have no clue on a potential transition to a sessile behavior during biofilm formation and its role, if any, in the efficacy of biofilm formation and enrichment in 3→3 cross-links. We predict that the content in 3→3 cross-link may also be increased in the stationary phase of growth, as it is the case in *E. coli*. Our study focuses on growth in two conditions, *i.e.* biofilm and planktonic. The discussion of these two topics is therefore beyond the scope of the current manuscript.

10. Methods section. For each assay indicate the number of technical and biological replicates.

Answer: The number of assay was indicated Lines 402, 438, 447, 451, and 459 of the revised manuscript.

Minor comments

1. L33. LdtPae3?

Answer: The test was modified as suggested (Line 33 of the revised manuscript)

2. L185. Indicate that 4 and 6 muopeptides were those identified in Fig 2 for clarity.

Answer: The text was modified as suggested (Line 170 of the revised manuscript).

3. L187. What does 'PF03734' refer to?

Answer: We have indicated that this ID number refers to a protein family domain (Line 73).

4. L189 - 190. Can authors include the PA14 locus tag for each ldt?

Answer: The locus tags were indicated in the revised version of the manuscript (Line 176).

5. Figure 3A is somewhat confusing. Are the reactions catalyzed by Ldt1 that indicated in red text, or where the navy circle is indicated? Expanding the figure legend to provide more detail would be helpful.

Answer: More details on the reaction catalyzed by Ldt_{Pae1} have been provided in the legend to the Fig. 3 (Lines 508 to 511). These reactions are described in Fig. 3A in a manner that is commonly used both for PBPs and LDTs.

6. L231 - 234. Refer to Fig2B here.

The appropriate figure was introduced as requested (Line 217).

7. L292. Specify that the 11.3% dimers were detected in biofilms, for clarity.

Answer: The text was modified as suggested (Line 283).

8. Figure 5D. * are mentioned in the figure legend, but there are none on the graph.

Answer: The figure was modified to incorporate stars.

April 18, 2023

Dr. Michel Arthur
INSERM
Paris
France

Re: Spectrum05217-22R1 (Characterization of *Pseudomonas aeruginosa* L,D-transpeptidases and evaluation of their role in peptidoglycan adaptation to biofilm growth)

Dear Dr. Arthur:

Thank you for submitting your revised manuscript, and for taking into account all of the reviewer's comments. I am pleased to accept your manuscript, and I am forwarding it to the ASM Journals Department for publication. You will be notified when your proofs are ready to be viewed.

Sincerely,

Eric Cascales
Editor, Microbiology Spectrum
